# In-Silico Identification of Novel Pharmacological Synergisms: The Trabectedin Case

**DOI:** 10.3390/ijms25042059

**Published:** 2024-02-08

**Authors:** Laura Mannarino, Nicholas Ravasio, Maurizio D’Incalci, Sergio Marchini, Marco Masseroli

**Affiliations:** 1Department of Biomedical Sciences, Humanitas University, Via Rita Levi Montalcini 4, Pieve Emanuele, 20072 Milan, Italy; maurizio.dincalci@hunimed.eu; 2Laboratory of Cancer Pharmacology, IRCCS Humanitas Research Hospital, Via Manzoni 56, Rozzano, 20089 Milan, Italy; sergio.marchini@humanitasresearch.it; 3Dipartimento di Elettronica, Informazione e Bioingegneria, Politecnico di Milano, 20133 Milan, Italy; nicholas.ravasio@outlook.it (N.R.); marco.masseroli@polimi.it (M.M.)

**Keywords:** in-silico drug repositioning, drug combination, trabectedin, lurbinectedin, drug synergism

## Abstract

The in-silico strategy of identifying novel uses for already existing drugs, known as drug repositioning, has enhanced drug discovery. Previous studies have shown a positive correlation between expression changes induced by the anticancer agent trabectedin and those caused by irinotecan, a topoisomerase I inhibitor. Leveraging the availability of transcriptional datasets, we developed a general in-silico drug-repositioning approach that we applied to investigate novel trabectedin synergisms. We set a workflow allowing the identification of genes selectively modulated by a drug and possible novel drug interactions. To show its effectiveness, we selected trabectedin as a case-study drug. We retrieved eight transcriptional cancer datasets including controls and samples treated with trabectedin or its analog lurbinectedin. We compared gene signature associated with each dataset to the 476,251 signatures from the Connectivity Map database. The most significant connections referred to mitomycin-c, topoisomerase II inhibitors, a PKC inhibitor, a Chk1 inhibitor, an antifungal agent, and an antagonist of the glutamate receptor. Genes coherently modulated by the drugs were involved in cell cycle, PPARalpha, and Rho GTPases pathways. Our in-silico approach for drug synergism identification showed that trabectedin modulates specific pathways that are shared with other drugs, suggesting possible synergisms.

## 1. Introduction

The long duration of time required for a new drug to be placed on the market, with related cost-intensive investments and the high risk of failure from the development to the clinical application, pushes toward the implementation of faster strategies for the identification of novel compounds [1,2]. Information technology and data mining have made this possible through the development of in-silico approaches, like drug repositioning, which refers to drugs that have a different indication from the one they were initially designed for. The advantage of this approach is that repurposing already existing drugs reduces considerably the time to market and the costs and somehow prevents risks since it deals with compounds whose potential side effects are already known [1,2,3].

Our study is situated in this context and has two main objectives with both a methodological and a biological nature: (1) defining a general methodological workflow for the identification of genes selectively modulated by drugs of interest and for the discovery of potentially new combinations; (2) taking the drug trabectedin as an example use case to show validity and usefulness of the defined workflow, applying it to find genes that specifically characterize its mechanism, and to identify drugs inducing similar transcriptional profiles, which could be suitable for the exploration of new combinatorial therapies.

The choice of trabectedin for a case study was dictated by the long-lasting experience of our laboratory in the study of this drug of marine origin, which has an extraordinarily complex and interesting mechanism [4,5]. It acts as a transcriptional modulator, whose efficacy depends on a functional DNA repair pathway, and it also has a specific effect on the tumor microenvironment [5]. Probably due to its pleiotropic MoA, trabectedin has been demonstrated to be a good candidate for drug combinations. It has been used in combination with a topoisomerase I inhibitor, like irinotecan, for treating Ewing sarcoma [6,7] or in combination with anthracyclines for the therapy for soft-tissue sarcoma [8] and ovarian cancer [9,10]. Recently, a phase II study testing the combination with an anti-diabetic compound, known as pioglitazone, has started for patients affected with myxoid liposarcoma [11,12]. To overcome an important clinical side effect of trabectedin, i.e., liver toxicity, an analog drug called lurbinectedin has been developed and introduced in the clinic [13]. Lurbinectedin contains the same pentacyclic skeleton of the fused tetrahydroisoquinoline rings as trabectedin, but the additional tetrahydroisoquinoline of trabectedin is replaced by a tetrahydro β-carboline. Given their similar structure, the two drugs present an overlapping MoA, and, at the molecular level, they induce similar transcriptional effects [14,15]. Interestingly, in our previous work, we found and confirmed the synergy between trabectedin and irinotecan through a drug repositioning approach [16].

One attractive drug repositioning procedure is the “guilt by association” principle. This approach assumes that if two drugs elicit a similar transcriptional profile, they are likely to share the same mode of action (MoA), independently of the biological system (cell line or tissue) they have been applied to [3,17]. This strategy has proven to be effective in finding connections between drugs that have also been successful in the clinic [18]. The reasons for looking for drugs with similar transcriptional responses are multiple: (1) A compound can improve the effectiveness of another one. (2) Given an enhanced response, the administration dose can be modulated. (3) Consequently, to the first two points, dose modulation can help in lowering the side effects. One of the already available implementations for drug repositioning is the Connectivity Map (CMap) [19,20], which offers a database of 476,251 expression profiles derived from the treatment of different cell lines with 27,927 molecules. CMap aims to find connections among cells and drug treatments and allows searching for synergisms through the “guilt by association” principle.

Guided by these premises, we hypothesized that trabectedin and its analog lurbinectedin could have a selective action on specific genes in different cellular systems for which some sensitivities to the drugs can be exploited through drug repositioning approaches.

## 2. Results

### 2.1. Workflow for Drug-Specific Gene Selection

To identify genes that are selectively modulated by a drug of interest, we set up a workflow that allows determining drug-specific genes starting from public or in-house datasets. Figure 1 shows the workflow, which is divided into five main steps.

Briefly, the first step (A) consists of extracting a gene signature for each dataset of interest using up- and down-regulated differentially expressed genes. In step B, the signatures are loaded on the clue.io platform, which provides functions for signature–signature comparison using the Connectivity Map database. Given the interest in the transcriptional effects of drugs, the analysis is restricted to the compounds only. In step C, four identifiers (pert_id, pert_iname, cell_id, and gene_info) are used to extract genes and their expression values, expressed as z-score, from the CMap database in gctx format. Genes with their expression values are then compared with each signature from step A to identify common genes with the same direction of transcriptional regulation to obtain a gene list (step D). Finally (step E), the extracted gene lists are evaluated through pathway analysis, to identify their associated relevant pathways and are further investigated with correlation and clustering analysis to identify similarities between drugs. The following sections show in detail the results obtained from each step of the defined workflow when applied to the considered case of trabectedin.

### 2.2. Selection of Transcriptional Datasets

We applied our workflow to a specific drug of interest, namely trabectedin and its analog lurbinectedin, which share the same mechanism of action. First, we defined the whole set of data useful to investigate the transcriptional effects of trabectedin/lurbinectedin. We used data generated with the same type of technology (i.e., microarray platforms), with the availability of treated vs. untreated samples that allow for identifying transcriptionally modulated genes upon drug exposure. We selected both tumor and non-tumor datasets from the ArrayExpress [21] or Gene Expression Omnibus (GEO) [22] public databases, as listed in Table 1. They encompass cell line models of different tumors, like MV411 (myelomonocytic leukemia), JN-DSCRT1 (desmoplastic sarcoma), OCILy7 (lymphoma), U2932 (lymphoma), SHP-77 (small cell lung cancer), immune systems cells (monocytes), and two patient-derived xenograft models of myxoid liposarcoma (PDX-II and PDX-III).

### 2.3. Extraction of Gene Signatures

The main transcriptional effects of a drug can be best attributed to the genes that show the highest response upon treatment. In CMap, these genes can be loaded as a query gene list of the most modulated genes under the considered conditions. Thus, for each dataset, we extracted a list of the first 150 most up-regulated and the first 150 most down-regulated genes, when present, and drew a dataset-specific gene signature for each considered dataset. All signatures are reported in Appendix A.

### 2.4. Identification of Compounds Correlated with the Drug of Interest

According to our defined workflow, the signature–signature comparison was performed using the CMap tool available on the clue.io platform. The *only compound selection* mode allows for selecting the compounds most connected to the gene expression profiles of the drug of interest (in our case trabectedin/lurbinectedin). In this way, in the CMap database, we found 10 compounds having high connections with trabectedin or lurbinectedin, namely the topoisomerase I inhibitor irinotecan or its active metabolite SN-38, topoisomerase II inhibitors (amsacrine and teniposide), mitomycin-c which binds to the DNA minor groove, SB-218078 which is an inhibitor of Chk1 kinase, SIB-1893 which is an antagonist of glutamate receptor, importazole which is an inhibitor of importin-Beta, bisindolylmalemide which is a protein kinase inhibitor, and pyrvinium-pamoate which is an anti-helminthync agent that inhibits Wnt pathway, as shown in the heatmap in Figure 2.

The heatmap color indicates the tau score, ranging from −100 (blue) to +100 (red), representing the value of the connection calculated between the query *signature* and each compound–cell line pair. For a better comprehension of the results, we reported the pre-identified *signatures* in the first line, depicted with colors as in the legend, while in the second line we showed the CMap cell line, again reported with colors as in the legend. Except for the pairs PDX-II-ET-24h1dose or PDX-II-ET-24h3dose and bisindolylmaleimide or SB-218078, overall, the trabectedin/lurbinectedin-induced *signatures* showed a high connection with all 10 compounds, independently on the cell line of origin. Heatmaps specific to each dataset are reported in Appendix A.

### 2.5. Identification of Genes with the Same Transcriptional Modulation

Once the highly connected compounds were identified, the investigation was moved to a deeper level. Since the connections of CMap are derived from genes having the same activation or inhibition following drug treatment, we aimed at identifying those genes coherently modulated by the drug of interest (in our case, both trabectedin/lurbinectedin) and each identified compound. Thus, from the GEO GSE92742 CMap data, we selected the genes and their transcriptional values from all the cell line–compound pairs previously found. These were compared to the genes of the query *signatures* and only common genes with the same direction of regulation (up or down) in at least 70% of all the samples were selected (Appendix A). To improve the investigation of the obtained common gene profiles, we checked whether the query signatures presented similarities among each other based on the selected genes. Thus, we made their clustering analysis (as specified in the Methods section), which identified two distinct groups: one composed of the JNDSCRT1, PDX-II, PDX-III, and MV411 datasets, and another one consisting of the U2932, OCILy7, SHP-77, and monocytes datasets (depicted in orange and green, respectively, in Appendix A). Each pair of signatures taken from the two groups presented a high correlation within the same group and less with the other one (Appendix A). Through this approach, we improved the investigation of the obtained common gene profiles and identified two main groups explaining the transcriptional similarities of the initial datasets.

### 2.6. Common Pathways

Gene lists can be meaningless if not placed in the biological context. Thus, a pathway analysis of the identified common genes was performed. As shown in Figure 3, in our case, we found 12 pathways: the Cyclin A/B1/B2 associated events during the G2/M transition pathway was associated with the activity of 8 out of 10 compounds, as expected from compounds of the same drug family (i.e., topoisomerase inhibitors); SN-38, irinotecan, and teniposide had the most pathways in common (7 out of 12), while the mitomycin-c and SB-218078 compounds showed the involvement of the PPARalpha pathway in the regulation of transcription.

The considered trabectedin use case demonstrates that the workflow we defined enables the identification of drugs inducing transcriptional effects similar to a specific drug treatment of interest. Indeed, the approach can be applied for the study of any drug.

## 3. Discussion

We developed a workflow for identifying drugs with similarities in their mechanism of action, taking advantage of the CMap database, and extracting and annotating coherently modulated genes. It can be easily applied to any drug of interest. As an example of a relevant application in our pipeline, we selected trabectedin and its analog lurbinectedin as candidate drugs for such a study, since these two drugs share similar and overlapping transcriptional profiles [14,15].

To work with similar data sources, we selected gene expression profiles from microarray technology, used to assess the transcriptional effect elicited by either trabectedin or lurbinectedin with respect to a control condition. Overall, we worked with datasets from monocytes of healthy donors or preclinical models of different tumors, like myxoid liposarcoma, desmoplastic sarcoma, lymphoma, and small-cell lung cancer.

The drug-induced transcriptional profiles were used to build gene signatures, i.e., lists of the most modulated genes sorted based on their differential expression; these were used as inputs for the CMap software (https://clue.io, accessed on 1 August 2020) [19] to identify drug–gene interactions. The use of these signatures of most modulated genes allowed us to consider each dataset as independent, thus overcoming any technical issue in comparing gene profiles from different sources and platforms while preserving the biological information on the drug’s effect. The method we used to select drugs with a similar mechanism of action was the so-called “guilt by association” principle as follows: if two drugs modulate the same genes in the same way, they are likely to share their MoA [1,3]. Thus, we focused on the transcriptionally modulated gene profiles rather than synergisms guided by the same molecular drug structure. Guided by this assumption, we used CMap to select the 10 most connected, e.g., correlated, drugs to trabectedin/lurbinectedin. The choice of using only 10 compounds was guided by the intent to focus on the closest drugs; however, the number of considered compounds can be freely chosen according to the specific analytical needs. Even though trabectedin and lurbinectedin are structurally very different from the other compounds for which a potential synergism is envisaged, they share the high lipophilicity of their active metabolites.

The workflow we propose in this work represents a step forward for the selection of genes that support the high connection between drugs. To this aim, we used the GSE92742 data from the CMap database. This dataset provides the genes modulated by all the compounds of the CMap database with their associated differential transcriptional value, expressed as a z-score, facilitating the identification of those genes that are up-regulated (positive z-scores) or down-regulated (negative z-scores) by a drug. Moreover, the GSE92742 data supplies useful metadata about compounds and cell lines, which are key to precisely identifying the datasets of interest. By comparing trabectedin/lurbinectedin signatures with the GSE92742 gene profiles and requiring only common genes with the same regulation direction (up or down) in at least 70% of all the samples, we were able to gather all the genes that support the high connections with trabectedin/lurbinectedin for each of the 10 most connected compounds.

Finally, we questioned whether the identified common genes could be linked to specific biological characteristics (i.e., annotations). Performing enrichment analysis, we showed that the main pathways under trabectedin/lurbinectedin MoA are related to the cell cycle, especially in the blockage of the G2/M phase. The cell cycle blockage in this phase is a hallmark of trabectedin [4,5,23]. It has been described in prostate cancer stem cells [24], and with an enhanced effect when trabectedin is administered in combination with other drugs like PARP1 inhibitors in sarcoma [25] or campthotecin in sarcoma models [26]. The arrest in the G2/M has also been connected to the other 8 out of the 10 compounds evaluated in this study. The other important biological function shared by trabectedin and the other identified compounds, especially mitomycin-c and SB-218078, is the transcriptional regulation guided by PPARalpha, a pathway that is already possible to target with PPARalpha-agonists, for example in the treatment of metabolic syndromes [27].

Among the obtained results, we found a high connection between trabectedin/lurbinectedin and the class of topoisomerase-I inhibitors. The efficacy of the combinatorial effect of trabectedin and one of these compounds, named irinotecan, was already validated in preclinical models of rhabdomyosarcoma [28] and in the clinic for the management of Ewing sarcoma [6,7]. The combinatorial administration of trabectedin and irinotecan already validated in the clinical setting further supports the potential of our approach.

Overall, we showed good results obtained by the application of our workflow. These prove the validity of the workflow we defined, which can be straightforwardly generalized to any drug of interest for which the search for novel synergisms is required. The identification of drugs with similar MoA opens new hypotheses for the exploration of novel combinatorial drug therapies.

## 4. Materials and Methods

### 4.1. Dataset

The used datasets were extracted from ArrayExpress [21] or Gene Expression Omnibus [22] according to the following criteria:Availability of control samples under untreated conditions;Availability of samples treated with either trabectedin or lurbinectedin;At least three samples for each condition (either treated or untreated);Gene expression data generated by microarray technology.

The final eight selected datasets are reported in Table 1 and further detailed in Appendix A.

### 4.2. Microarray Data Analysis

Each dataset was analyzed independently. Microarray data analysis was performed as previously reported [16], starting from normalized data when available (all datasets, except for OCLy7 and U2932); otherwise, the R package Limma version 3.46.0 [29] was used to normalize the data with the “quantile” normalization method [30]. All datasets were annotated according to the corresponding microarray source and annotation package, as reported in Table 2. The R package Limma version 3.46.0 [29] was used to identify differentially expressed genes (DEGs) by comparing treated conditions to controls. DEGs were filtered with a q-value of 0.05.

### 4.3. Connectivity Map

The Connectivity Map resource was accessed through the clue.io platform (https://clue.io, accessed on 1 August 2020) to identify drugs that elicit a similar response to that of trabectedin or lurbinectedin based on differential gene expression signatures. CMap allows for finding connections between diseases, genes, and pharmacological compounds.

### 4.4. Definition of a Gene Signature

The identified DEGs were sorted in descending order according to their base-two logarithmic-fold change (logFC). Since CMap allows queries with a maximum of 300 tags (i.e., genes), the first 150 most up-regulated and the first 150 most down-regulated DEGs for each dataset were selected when present. Only genes belonging to the BING (Best INferred Genes) [31] list of the clue.io platform were retained. For each dataset, these lists of DEGs were defined as gene signatures using the *listmaker* function of clue.io.

### 4.5. CMap Query

The clue.io platform provides different functions for the upload and analysis of data. All eight created gene signatures were loaded through the *query* function and used to identify compounds with similar expression profiles, which were then visualized with the *heatmap* function. Given the interest in searching for drugs with a similar mechanism of action following the “guilt by association” principle [3,17], results were limited to the compounds. They were sorted from the most to the least expression–correlated ones, based on the tau score. This last one, also known as “the CMap connectivity score”, is defined as the fraction of reference gene sets (originating from the signatures present in Touchstone, the reference database of CMap) with greater similarity to the compound than that of the gene signature under query, as reported in the clue.io documentation [31]. The tau score ranges from −100 to +100; when the score is higher (or lower) than +90 (or −90), the result is considered worthy of further investigation, and the CMap query, i.e., the loaded gene signature, can be defined as “connected” to that result, i.e., compound. All results were saved in .gctx file format [32] for further processing.

### 4.6. CMap Query Result Selection

The results in .gctx format were parsed with iPython [33] through a GCTx parser of cmapPy [32] imported with the specific Python instruction from cmapPy.pandasGEXpress.parse import parser. The first 10 results were selected according to their tau score. Graphic representations through heatmaps were created with the *seaborn* version v0.101 package using the *clustermap* function [34].

### 4.7. Identification of Genes with the Same Up- or Down-Regulation Direction

The selection of genes with commonalities between each dataset and the CMap database, and with the same up or down direction of regulation by the treatment, was conducted as follows.

Gene expression data related to the CMap dataset were retrieved from the Gene Expression Omnibus—GEO (https://www.ncbi.nlm.nih.gov/geo, accessed on 20 July 2020). CMap datasets are based on Phase I data [20], which are stored in GEO as Series GSE92742; from them, we specifically used the following:GSE92742_Broad_LINCS_sig_info.txt, to retrieve information on cell lines and compounds;GSE92742_Broad_LINCS_gene_info.txt, to retrieve information on genes;GSE92742_Broad_LINCS_Level5_COMPZ.MODZ_n473647x12328.gctx, representing the level 5 data z-score vector [19].

The Python *cmapPy* package, version 4.0.1 [32] was used to parse the GSE92742 gctx file. The gctx format is an evolution of the gct text file, specifically for the storage of large matrices associated with metadata annotations [32].

The sig_info file (GSE92742_Broad_LINCS_sig_info.txt) was used to subset the previous gctx file based on the following:The pertubagen (i.e., compound) id (pert_id), using the pert_id corresponding to the first 10 results selected as previously reported in section “CMap query result selection”, as reported in Table 3;The pertubagen name (pert_iname), using the pert_iname corresponding to the first 10 results selected as previously reported in section “CMap query result selection”, as reported in Table 3;The cell lines of human origin of interest (cell_id) associated with the CMap database, as reported in Table 4.

The gene_info file (GSE92742_Broad_LINCS_gene_info.txt) was used to select only the 978 *landmark* genes defined as widely expressed across lineage, which have been determined through the L1000 assay according to the clue.io documentation [31].

The application of these filters resulted in the final file in gctx format. Then, each gene signature related to the datasets of the cell lines in Table 4 was divided into up- and down-regulated genes, which were used to subset the selected gctx file independently for each dataset. Only genes with the same up- or down-regulation direction in at least 70% of all considered samples were selected.

### 4.8. Correlation and Clustering

Pearson correlation between the gene signatures of each pair of datasets was conducted with the corr() function of the Python pandas package version 1.0.4 for data analysis using the default method of the Pearson correlation [35]. Dataset clustering was performed with the dendrogram and linkage functions from the sciPy library version 1.5.0 [36], using the Ward variance minimization algorithm.

### 4.9. Functional Analysis

Genes were annotated using the org.Hs.eg.db [37] database of human annotations. Common genes were analyzed for functional classification with the clusterProfiler version 3.18.1 R package [38], using both the groupGO function to query the gene ontology (molecular function—MF ontology and biological process—BP ontology) [39] and the enrichPathway function for enrichment analysis, based on hypergeometric test, with the Reactome database [40], using *p*-value cutoff = 0.05 and the multiple testing correction of *p*-value with the Benjamini–Hochberg procedure [41]. Pathway visualization was conducted with iPython [33] using the matplotlib version 3.2.2 [42] and the seaborn version v0.10.1 [34] packages.

### 4.10. Workflow Implementation

The defined workflow was implemented with an iPython notebook, publicly available at https://drive.google.com/drive/folders/1g2w2fOVH8ulAZ82PYR8SAT4fO3PPgf_3?usp=share_link, accessed on 5 February 2024.

## 5. Conclusions

The present work provides a workflow to investigate potential drug synergisms. The results presented here show that our workflow is useful for the discovery of drugs with a similar MoA and for the identification and pathway analysis of the genes that support the hypothesis of these pharmacological synergisms.

The merits of this work rely on (1) the definition and automation of an easily reproducible workflow applicable to the study of any drug; (2) the use of heterogeneous biological models, such as cell lines, patient-derived xenograft, and human blood cells, which help in the establishment of genes whose expression modulation is strictly drug-dependent; and (3) the identification of novel compounds that can potentially be used in synergy with trabectedin or its analog lurbinectedin, like mitomycin-c, amsacrine, bisindolylmaleimide, pyrivinium-pamoate, SIB-1893, and SB-218078, which were never proposed before as such.

Being an in-silico approach, our work has the limitation of lacking experimental validation. The strength of the results obtained through these approaches strongly relies on the following confirmation with laboratory assessments. Indeed, when computationally computed drugs advance to the clinical stage, their chances of failure are notably high [43]. The causes can be identified by the use of too-simple models, like the cell lines in the CMap database, that are far from recapitulating the complexity of biological systems and make it difficult to generalize results to the real population. On the other hand, it is worth highlighting that in-silico methods do not presume to provide ready-to-use solutions; instead, they are positioned in the initial stages of the experimental process aimed at quickly generating a hypothesis.

Given the previous considerations, as a future extension of the presented work, experimental biological testing is needed to assess and finally validate the relevant pharmacological results obtained in the example application of our workflow.

## Figures and Tables

**Figure 1 ijms-25-02059-f001:**
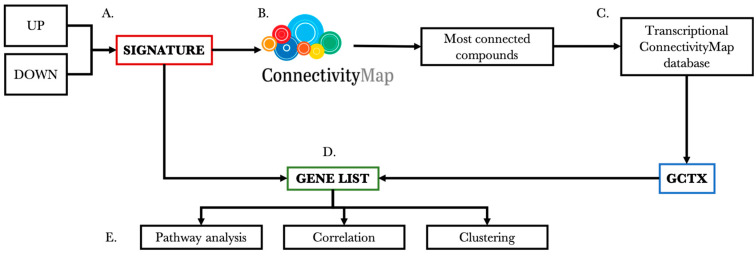
Schematic representation of the general workflow for the identification of genes selectively modulated by a drug of interest, which are defined and used in this work.

**Figure 2 ijms-25-02059-f002:**
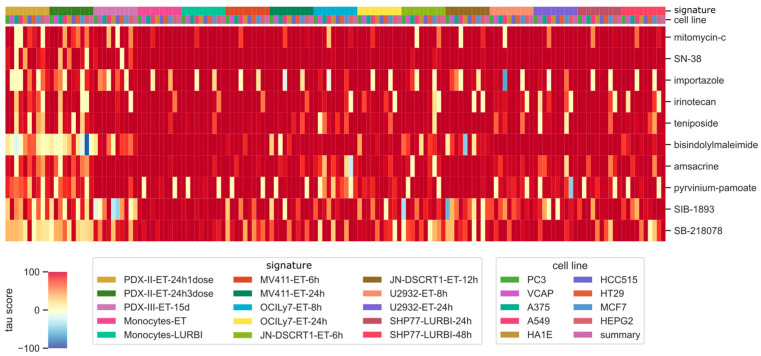
Heatmap showing the first 10 compounds with the highest connection with trabectedin/lurbinectedin gene expression profiles. The names of the compounds are on the right and listed top-down from the most to the least connected one. The colors of the heatmap represent the tau score, ranging from −100 (low connection) to +100 (high connection). The darker the color the higher the connection: reds for positive values, blue for negative values. Lines on the top of the heatmap: the first represents the dataset signature as reported in the legend “signature”; the second represents the cell line as reported in the legend “cell line”.

**Figure 3 ijms-25-02059-f003:**
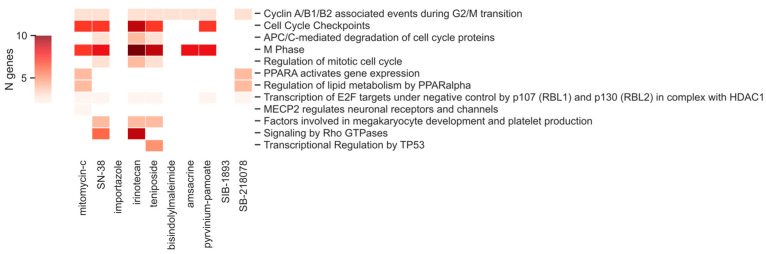
The figure shows on the x-axis the compounds analyzed in this work and on the y-axis the corresponding pathways identified as significant by the pathway analysis. Colored rectangles mean the significance of the pathway for the considered compound–pathway pairs. Colors represent the number of genes in the pathway as reported in the legend on the left (N genes). White indicates no significant pathway.

**Table 1 ijms-25-02059-t001:** List of the selected datasets.

Dataset ID	Brief Description	Origin	Available at	Accession ID
MV411	Gene expression profiling of a cell line model of myelomonocytic leukemia	Cell line	ArrayExpress	E-MTAB-2978
JN-DSCRT1	Desmoplastic small round cell sarcoma	Cell line	ArrayExpress	E-MTAB-4532
PDX-II	Myxoid liposarcoma—type II	Patient-derived xenograft	ArrayExpress	E-MTAB-8632
PDX-III	Myxoid liposarcoma type III	Patient-derived xenograft	ArrayExpress	E-MTAB-8632
OCILy7	Lymphoma DLBCL subtype germinal center B cell (GBC)	Cell line	Gene Expression Omnibus	GSE104197
U2932	Lymphoma DLBCL subtype activated B cell-like (ABC)	Cell line	Gene Expression Omnibus	GSE104197
Monocytes	Immune system cells from donors	Blood cells	ArrayExpress	E-MTAB-5366
SHP-77	Small cell lung cancer	Cell line	in-house	-

**Table 2 ijms-25-02059-t002:** Annotation packages and microarray sources of the used datasets.

Dataset ID	Annotation Package	Source
MV4-11	HsAgilentDesign026652	Agilent, Santa Clara, CA, USA
JNDSCRT1	HsAgilentDesign039494	Agilent, Santa Clara, CA, USA
PDX-II, PDX-III	HsAgilentDesign026652	Agilent, Santa Clara, CA, USA
OCILy7, U2932	IlluminaHumanv4	Illumina, San Diego, CA, USA
Monocytes	HsAgilentDesign039494	Agilent, Santa Clara, CA, USA
SHP-77	HsAgilentDesign039494	Agilent, Santa Clara, CA, USA

**Table 3 ijms-25-02059-t003:** The unique compound identifier (pert_id) and name (pert_iname) of each considered compound.

pert_id	part_iname
BRD-A02481876	importazole
BRD-A06352508	SB-218078
BRD-A35588707	teniposide
BRD-A36630025	SN-38
BRD-A48237631	mitomycin-c
BRD-K08547377	irinotecan
BRD-K31342827	bisindolylmaleimide
BRD-K67439147	SIB-1893
BRD-K98490050	amsacrine
BRD-M86331534	pyrvinium-pamoate

**Table 4 ijms-25-02059-t004:** Considered cell lines of the CMap database.

cell_id	Brief Description
A375	malignant melanoma
A549	non-small cell lung carcinoma
HA1E	kidney epithelial immortalized (non-cancer cell line)
HCC515	non-small cell lung adenocarcinoma
HEPG2	hepatocellular carcinoma
HT29	colorectal adenocarcinoma
MCF7	breast adenocarcinoma
PC3	prostate adenocarcinoma
VCAP	metastatic prostate cancer

## Data Availability

At https://drive.google.com/drive/folders/1g2w2fOVH8ulAZ82PYR8SAT4fO3PPgf_3?usp=share_link, accessed on 5 February 2024, we provide the defined workflow implemented within iPython notebooks.

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
