# Peer review of "In-Silico Identification of Novel Pharmacological Synergisms: The Trabectedin Case"

_ijms, 2024, doi:10.3390/ijms25042059_

Round 1

Reviewer 1 Report

Comments and Suggestions for Authors

·    The introduction provides good background on drug repositioning and the rationale for studying trabectedin. However, the specific objectives could be stated more clearly upfront. Consider moving the given objectives currently at the end of the introduction to the beginning.

·    The results are clearly presented following the workflow steps. However, some additional interpretation of the biological relevance of the findings would strengthen this section. For example, when discussing the common pathways, expand on why modulation of certain pathways like cell cycle and PPARalpha are relevant to the mechanism of trabectedin.

·    There are some grammar issues throughout that should be addressed, such as missing articles ("allows identification of genes" should be "allows the identification of genes") and sentence fragments. Carefully proofread the manuscript.

·    The discussion could be expanded to provide more context for the results and how they advance understanding of trabectedin's mechanism and potential combinations. Compare the synergisms identified here to previously known combinations with trabectedin to highlight novelty.

·    The limitations of the in-silico approach should also be acknowledged in the discussion. Comment on any experimental validations or future directions to build on these computational findings.

Author Response

Q1) The introduction provides good background on drug repositioning and the rationale for studying trabectedin. However, the specific objectives could be stated more clearly upfront. Consider moving the given objectives currently at the end of the introduction to the beginning.

A1) We thank the reviewer for this proposed advancement for the introduction section. As suggested, we moved the objectives of our study to the beginning of the Introduction (pages 1-2, lines 41-47) and rephrased the entire section accordingly.

Q2) The results are clearly presented following the workflow steps. However, some additional interpretation of the biological relevance of the findings would strengthen this section. For example, when discussing the common pathways, expand on why modulation of certain pathways like cell cycle and PPARalpha are relevant to the mechanism of trabectedin.

A2) We thank the reviewer for this point. We have discussed further the importance of these two pathways in the Discussion (page 7, lines 310-317) and added further supporting references.

Q3) There are some grammar issues throughout that should be addressed, such as missing articles ("allows identification of genes" should be "allows the identification of genes") and sentence fragments. Carefully proofread the manuscript.

A3) We apologize for this inconvenience. We have revised the grammar and corrected the issues like missing articles and misspellings throughout the text.

Q4) The discussion could be expanded to provide more context for the results and how they advance understanding of trabectedin's mechanism and potential combinations. Compare the synergisms identified here to previously known combinations with trabectedin to highlight novelty.

A4) We thank the reviewer for this important observation. We have further discussed this point in the Discussion (page 7, lines 322-323).

Q5) The limitations of the in-silico approach should also be acknowledged in the discussion. Comment on any experimental validations or future directions to build on these computational findings.

A5) We thank the reviewer for stressing this point. We agree that the lack of experimental validation represents a limitation to our work. To address this point, we have discussed this in the Conclusions section (page 10, lines 452-461).

Reviewer 2 Report

Comments and Suggestions for Authors

This manuscript provides a workflow to investigate potential drug synergisms. By selecting drug-specific gene, selecting transcriptional dataset, extraction of gene signatures, identification of compounds correlated with the drug of interest, identification of genes with the same transcriptional modulation, and finding common pathways, the workflow reported here is useful for the discovery of drugs with a similar MoA and for the identification and pathway analysis of the genes that support the hypothesis of these pharmacological synergisms.

However, all these results are based on in-silico, and experimental biological testing is needed to assess and finally validate the relevant pharmacological results obtained in the example application of this workflow for further study.

Author Response

This manuscript provides a workflow to investigate potential drug synergisms. By selecting drug-specific gene, selecting transcriptional dataset, extraction of gene signatures, identification of compounds correlated with the drug of interest, identification of genes with the same transcriptional modulation, and finding common pathways, the workflow reported here is useful for the discovery of drugs with a similar MoA and for the identification and pathway analysis of the genes that support the hypothesis of these pharmacological synergisms. 

Q1) However, all these results are based on in-silico, and experimental biological testing is needed to assess and finally validate the relevant pharmacological results obtained in the example application of this workflow for further study.

A1) We thank the reviewer for this consideration with which we fully agree. In-silico approaches have strong limitations related to the lack of experimental or pharmacological validation. Given the importance of this point, we have added a paragraph in the Conclusions section (page 10, lines 452-461) discussing the limitations of our approach.

Reviewer 3 Report

Comments and Suggestions for Authors

To the authors,

In this work by Mannarino et al., mined transcriptional datasets to investigate plausible synergism of trabectedin/lurbinectedin with other established drugs.  Cross referencing with the Connectivity Map database, they were able to identify synergisms with molecules targeting diverse pathways - cell cycle, PPARa, and Rho GTPases pathways.  The results do appear to back up the authors conclusions. However, I have some minor concerns that are as follows:

Major:

1.        In figure 3, where the authors depict the compounds analyzed and the associated significant pathways. However, the statistics (p values) for the “significant” observations is missing. Can the authors please clarify if a statistical analysis was performed to assess the significance of the pathways. If not, please substitute the word “significant” with a suitable word. 

Minor:

1.        The authors can depict the molecular structures of trabectedin/lurbinectedin and the hits from their study to give a better structural overview of compounds that have a putative synergistic effect with trabectedin.

2.        Inconsistent formatting of references- In the references section, a complete overhaul in formatting is needed. In some cases, the Journal name has been italicized but the abbreviation lacks punctuation (dots), while for some the same journal name shows up in its unabbreviated form.

a.        For example, reference 8 (and many others) have abbreviated and italicized but non-punctuated journal names, British Journal of Cancer has been abbreviated as Br J Cancer, while abbreviations normally appear as Br. J. Cancer

b.        References 18, 20, 21 (and many others) shows the full journal name (must be abbreviated and italicized).

Comments on the Quality of English Language

Minor formatting required. English language fine.

Author Response

To the authors,

In this work by Mannarino et al., mined transcriptional datasets to investigate plausible synergism of trabectedin/lurbinectedin with other established drugs.  Cross referencing with the Connectivity Map database, they were able to identify synergisms with molecules targeting diverse pathways - cell cycle, PPARa, and Rho GTPases pathways.  The results do appear to back up the authors conclusions. However, I have some minor concerns that are as follows:

Major:

Q1) In figure 3, where the authors depict the compounds analyzed and the associated significant pathways. However, the statistics (p values) for the “significant” observations is missing. Can the authors please clarify if a statistical analysis was performed to assess the significance of the pathways. If not, please substitute the word “significant” with a suitable word. 

A1) We apologize for the lack of clarity. All the pathways colored in red in Figure 3 are significant. To clarify this, we modified the legend of Figure 3 (page 6, line 256) and also added a clarification in the Materials and Methods section (page 10, lines 431-433).

Minor:

Q2) The authors can depict the molecular structures of trabectedin/lurbinectedin and the hits from their study to give a better structural overview of compounds that have a putative synergistic effect with trabectedin.

A2) We thank the reviewer for this point that focuses on the drug synergism explained by the molecular structures of the compounds. Even though trabectedin and its analog lurbinectedin are structurally very different from the other compounds that we have defined as most connected based on the gene expression profiles, they share with them the high lipophilicity of their active metabolites. We have added this observation in the Discussion section (page 6, lines 283-284 and lines 288-291).

Q3) Inconsistent formatting of references- In the references section, a complete overhaul in formatting is needed. In some cases, the Journal name has been italicized but the abbreviation lacks punctuation (dots), while for some the same journal name shows up in its unabbreviated form. a. For example, reference 8 (and many others) have abbreviated and italicized but non-punctuated journal names, British Journal of Cancer has been abbreviated as Br J Cancer, while abbreviations normally appear as Br. J. Cancer b. References 18, 20, 21 (and many others) shows the full journal name (must be abbreviated and italicized).

A3) We apologize for the inconvenience and thank the reviewer for highlighting this point. We have amended all the references as suggested (pages 11-13).

Reviewer 4 Report

Comments and Suggestions for Authors

The manuscript “In-silico identification of novel pharmacological synergisms: the trabectedin case” introduces an in-silico drug-repositioning approach using transcriptional datasets, focusing on trabectedin, an anticancer agent. The structure and content of the manuscript are well organized. The presented workflow is also comprehensive and demonstrates its utility in identifying potential drug synergisms. However, there are some minor aspects that could be considered for an improvement of the manuscript:
- The study relies heavily on in-silico analysis and computational predictions. Including experimental validation of identified drug synergisms would strengthen the credibility of the findings.
- It would be useful mentioning of additional details when using statistical methods (specific tests used or significance thresholds applied).
- the study mentions the use of different data sets. Did the authors identify potential biases or limitations associated with particular data sets? Addressing the variability of data sources and their impact on applicability could strengthen the study.
- The study mentions that the workflow is implemented in an iPython notebook. It is not mandatory, but making the workflow code and/or implementation publicly available, would facilitate transparency, reproducibility of the results

Author Response

The manuscript “In-silico identification of novel pharmacological synergisms: the trabectedin case” introduces an in-silico drug-repositioning approach using transcriptional datasets, focusing on trabectedin, an anticancer agent. The structure and content of the manuscript are well organized. The presented workflow is also comprehensive and demonstrates its utility in identifying potential drug synergisms.

However, there are some minor aspects that could be considered for an improvement of the manuscript:

Q1) The study relies heavily on in-silico analysis and computational predictions. Including experimental validation of identified drug synergisms would strengthen the credibility of the findings.

A1) We thank the reviewer for highlighting this point. We fully agree that the lack of experimental validation of in-silico results constitutes a limitation to our work. Thus, we have addressed this point in the Conclusions section of the study (page 10, lines 452-461)

Q2) It would be useful mentioning of additional details when using statistical methods (specific tests used or significance thresholds applied).

A2) We apologize for the lack of clarity. We have amended the Material and Methods adding more details: in section 4.8 “Correlation and clustering” we have specified the type of correlation used (pages 9-10, lines 422-423); in section 4.9 “Functional analysis” we have specified the type of enrichment test used for pathway analysis, and the multiple testing approach used (page 10, lines 431-433).

Q3) the study mentions the use of different data sets. Did the authors identify potential biases or limitations associated with particular data sets? Addressing the variability of data sources and their impact on applicability could strengthen the study.

A3) We thank the reviewer for this important point. In this work, we have used gene profiles from different sources, even if all are derived from the microarray technology. We are aware that this could impact results when different data sets are analyzed together, thus requiring approaches like normalization to make them comparable. However, in this study, we aimed to identify the most drug-modulated genes in each context considering the datasets alone as explained in the “Extraction of gene signatures” paragraph (page 4, lines 190-196). This allowed us to consider each data set independently. We discussed further this point in the Discussion section (page 6, lines 277-280).

Q4) The study mentions that the workflow is implemented in an iPython notebook. It is not mandatory, but making the workflow code and/or implementation publicly available, would facilitate transparency, reproducibility of the results

A4) We are in line with the reviewer since we sustain the reproducibility of scientific results. For this reason, the iPython notebooks that we have used in this work are publicly available as detailed in the Materials and Methods section 4.10 “Workflow implementation” (page 10, lines 436-438).